# Anaerobic Ammonium Oxidation Bacteria in a Freshwater Recirculating Pond Aquaculture System

**DOI:** 10.3390/ijerph18094941

**Published:** 2021-05-06

**Authors:** Xing-Guo Liu, Jie Wang, Zong-Fan Wu, Guo-Feng Cheng, Zhao-Jun Gu

**Affiliations:** 1Fishery Machinery and Instrument Research Institute, Chinese Academy of Fisheries Sciences, Shanghai 200092, China; wangjie@fmiri.ac.cn (J.W.); wuzongfan@fmiri.ac.cn (Z.-F.W.); chengguofeng@fmiri.ac.cn (G.-F.C.); guzhaojun@fmiri.ac.cn (Z.-J.G.); 2East China Normal University, Shanghai 200241, China

**Keywords:** anammox bacteria, environmental parameter, diversity, abundance, distribution, recirculating pond aquaculture system

## Abstract

Anaerobic ammonium oxidation (anammox) is a key biochemical process to reduce nitrogen pollution in aquaculture, especially in water recirculating pond aquaculture system (RPAS). We used 16S RNA and quantified PCR to study the distribution and environmental impacts of anammox bacteria in RPAS. The results show that the anammox bacterial community distributions and diversities that are apparently unit-specific and seasonal have significant (*p* < 0.05) difference variation in the RPAS. Most of the anaerobic ammonium oxidation bacteria sequences (77.72%) retrieved from the RPAS belong to the Brocadia cluster. The abundance of anammox bacterial in the RPAS ranged from 3.33 × 101 to 41.84 × 101 copies per ng of DNA. The environmental parameter of temperature and nitrogen composition in water could have impacted the anammox bacterial abundance. This study provides more information on our understanding of the anammox bacteria in the RPAS, and provides an important basis for RPAS improvement and regulation.

## 1. Introduction

Anaerobic ammonium oxidation (anammox) is a key reaction in RPAS, directly influencing the RPAS efficiency. Since anammox was first found in a wastewater treatment plant in the Netherlands in 1995 [1], widespread anammox bacteria have been found in marine environments, and there is growing evidence of its presence in estuaries [2,3], wetlands [4,5], freshwater rivers [6], freshwater lakes and various soil types [7,8,9]. However, only limited investigations regarding anammox bacteria have been conducted for aquaculture systems. Moreover, the reported works are mainly focused on indoor recirculating aquaculture systems (RAS) [10,11] and marine aquaculture ponds [12]. China is the largest aquaculture country in the world, and the proportion of aquaculture in fish production increased to 73.7% in 2016 [13]. However, since the 1980s, rapid deterioration by diseases has become frequent, pollution emissions have increased, and the production quality has declined, causing serious challenges to aquaculture [14]. A water recirculating pond aquaculture system (RPAS) is an advanced technology aquaculture around the world. RPAS could change traditional pond aquaculture and has the following advantages: (1) improves yield and production efficiency; (2) improves the digestibility and absorption rate of feed; (3) reduces production energy consumption and recycles used water; (4) improves labor efficiency and supports multi-level breeding, which can be well distributed to the market; and (5) is beneficial to the industrialized management of pond aquaculture. However, to the best of our knowledge, there is limited information available regarding anammox bacteria in freshwater RPAS. In this paper we detected the anammox process by phylogenetic and quantitative PCR approach to investigate the distribution, diversity and abundance of anammox bacteria in the RPAS and the relationship between their bacterial characteristics and the prevalent environmental factors, which are important for RPAS improvement and regulation.

## 2. Materials and Methods

### 2.1. RPAS Description

This study chose a typical water recirculating pond aquaculture system (RPAS) in China: the RPAS was located in Punan aquaculture farm (31°57′01″ N, 120°08′52″ E), Songjiang, Shanghai, China, and covers an area of about 2 ha. It consists of three functional units: an aquaculture pond (CP), pre-treatment unit (PT) and constructed wetland (CW) [15]. The RPAS system was started in 2010 (Figure 1).

### 2.2. Sample Collection and Physicochemical Analyses

Seven representative sample sites (black triangles in Figure 1b) were selected from pond 1# to CW and labeled with A–G. Sites A–C represent the culture pond unit (CP). Sites D and E are located at the eco-ditch and eco-pond, respectively, representing the pre-treatment unit (PT). Sites F and G represent the inlet and outlet parts of CW, respectively. In CP and PT units (sites A–E), surface sediments (0–5 cm) and corresponding overlying waters within a 0.5 m radius were collected. Three replicate sub-samples were collected at each sampling site and homogenized to obtain a representative sample. Due to the absence of soil in CW, samples of gravel from sites F and G were collected for the molecular analyses. For each site, the gravel sample was taken by layer at the depths of 0–20 (surface layer), 30–40 (middle layer) and 50–60 cm (bottom layers), and then were mixed. Interstitial water samples from sites F and G were also collected. All sediment, gravel and water samples were immediately put into ice boxes and transported back to the laboratory shortly after collection. In the laboratory, samples for subsequent molecular studies were kept at −80 °C and samples for physiochemical analysis were processed immediately. Sampling was conducted in March, June and September 2014, which correspondingly represented the prophase, metaphase and anaphase of the aquaculture period.

Temperature, pH, salinity, redox potentials (ORP) and dissolved oxygen (DO) in overlapping water of sites A–E and interstitial water in sites F and G were measured in situ prior to sampling using an YSI Pro Plus (YSI Inc., Yellow Springs, OH, USA). Inorganic N of the sediments was extracted with 2 M KCl in a 1:4 ratio (sediment to solution) for 1 h. Total phosphorus (TP), orthophosphate (PO_4_^+^-P), nitrite (NO_2_^−^-N), nitrate (NO_3_^−^-N), ammonia (NH_4_^+^-N), total nitrogen (TN) and total organic carbon (TOC) in water samples and PO_4_^+^-P, NO_2_^−^-N, NO_3_^−^-N, NH_4_^+^-N and TOC in sediment samples were quantified spectrophotometrically according to a previous study [16]. Sediment dry weights (DW) were measured after being dried in an oven at 105 °C for 24 h until constant weight was achieved. The concentration of PO_4_^+^-P, NO_2_^−^-N, NO_3_^−^-N, NH_4_^+^-N and TOC in sediments were recalculated based on their DW contents. All the tests were operated in three replicates.

### 2.3. DNA Extraction and Nested PCR

Environmental DNA was extracted from sediments samples using Power Soil DNA Kit (Mo Bio Laboratories, Inc., Carlsbad, CA, USA), following the manufacturer’s instructions. For the gravel samples, moderate amounts of the samples were mixed with 200 mL sterile PBS solution, and then rotated at 200 r/min for 1 h. The suspension solution was filtered with 0.22 μm Millipore Sterivex™ filter units (Millipore Corporation, Billerica, MA, USA). Genomic DNA was extracted from the filters using Power Soil DNA Kit. DNA of each sample was extracted three times and pooled to minimize the bias. Nested-PCR assay was conducted to detect anammox 16S rRNA genes, the detailed steps refer to the literature of Zhu et al. [17]. The initial amplification was carried out using the PLA46f-630r primer combination with a thermal profile of 96 °C for 10 min, followed by 35 cycles of 60 s at 96 °C, 1 min at 56 °C and 1 min at 72 °C [18]. Then, a 500-times diluted (1 μL) PCR product was used as template for the second amplification with Amx368f-Amx820r primers using a thermal profile of 96 °C for 10 min, followed by 25 cycles of 30 s at 96 °C, 1 min at 58 °C, 1 min at 72 °C [19]. All quantitative PCR products from different primer sets were electrophoresed on 1% agarose gels.

### 2.4. Quantitative PCR Assay

The 16S rRNA genes of anammox bacteria and total bacteria were analyzed by quantitative PCR (qPCR) using CFX96 real-time system (Life Science Research, Education, Process Separations, Food Science. Bio-Rad Laboratories (Shanghai) Co., Ltd., Room 601, Anlian Building, No. 168 Jingzhou Road, Yangpu District, Shanghai, China 200082). Anammox bacterial 16S rRNA genes were quantified with the primer set of AMX-808-F/AMX-1040-R and the TaqMan probe AMX-931 according to the TaqMan fluorogenic PCR method developed by Hamersley et al. [20]. The general bacterial primer pair of 1055F/1392R was selected for the total bacterial 16S rRNA gene assay [21], the specific steps were performed according to Zhu et al. [17]. All qPCR amplifications were performed in triplicate. Negative controls containing no template DNA were subjected to the same qPCR procedure to exclude any possible contamination.

Standard curves were constructed with ten-fold serial dilution of standard plasmids containing corresponding target 16S rRNA genes. The plasmid DNA concentration was determined on GeneQuant II RNA/DNA calculator (Amersham-Pharmacia Biotech., Buckinhamshire, UK), the copy number of target genes was calculated from the concentration of extracted plasmid DNA. The qPCR amplification efficiency was 90–93%, the correlation coefficients (R2) were greater than 0.99 for the two targeted genes.

### 2.5. Cloning, Sequencing and Phylogenetic Analysis

PCR-amplified DNA fragments of the desired size were excised from the gel and purified with GeneJET™ Gel Extraction Kit (Thermo Scientific, Waltham, MA, USA). The obtained purified fragments from sites A–C, sites D–E and sites F–G were equivalently pooled (m/m) to form 3 composite samples, representing the units of culture pond, pre-treatment unit and constructed wetland, respectively. Each of the composite DNA fragments was ligated into a pUCm-T vector (Sangon Biotech Co., Ltd., Shanghai, China) and transformed into Escherichia coli DH5a. Cloned inserts of targeting gene fragments were verified by PCR amplification with the primers M13 F/R. Sequencing was performed by Sangon Biotech Co., Ltd. (Shanghai, China). The obtained DNA sequences were examined and edited using DNASTAR Laser gene SeqMan Program (DNASTAR, Madison, WI, USA). The most closely related 16S rRNA gene sequences were found in the public databases NCBI BLAST (http://www.ncbi.nih.gov, accessed on 6 May 2021). The partial 16S rRNA genes of anammox bacteria were manually compiled and aligned by using Clustal W [22]. Mega4.5 adjacency method was used to construct phylogenetic tree, and 1000 repeated bootstrap resampling was used to estimate the confidence of tree nodes.

### 2.6. Nucleotide Sequence Accession Numbers

The sequences determined in this study for anammox bacteria are available in GenBank database under accession numbers KP701041, KP701042, KP701045 and KP729590–KP729598.

### 2.7. Statistical Analysis

Operational taxonomic units (OTUs) for community analysis were defined by a 3% difference in nucleotide sequences. The coverage, diversity (Shannon, Simpson) and richness index (Chaol and SACE) for each clone library were calculated by Mothur v1.33.3 program. The correlations between microbial communities and environmental factors were analyzed using Pearson’s moment correlation analysis of SPSS version 16.0. The redundancy analysis (RDA) of CANOCO (ver. 4.5; Microcomputer Power, Ithaca, NY, USA) was chosen to analyze the relationship between anammox community distribution and environmental factors [23].

## 3. Results

### 3.1. Physicochemical Characteristics of the RPAS

In eastern China, the pond culture cycle is generally about 10 months. In China pond culture, the total nitrogen (TN), total phosphorus (TP) and chemical oxygen demand are required to be less than 5 mg/L, 1 mg/L and 25 mg/L, respectively [24]. Therefore, the water quality of pond culture is also required to be within this range. During the 10 months of the experiment period, the physicochemical properties of the sampling sites in the RPAS are summarized in Table 1. All of the sampling sites showed slightly alkaline pH values between 7.02 and 7.65. For the TOC, TN, and NH_4_^+^-N concentrations in the water, the values ranked in the order of CP > PT > CW, implying the purifying effects of the PT and CW. The concentration of NH_4_^+^-N, NO_2_^−^-N and NO_3_^−^-N in the water of the system were relatively low, indicating its suitability for fish culture. For the sediments, the TOC contents were between 2.16 and 3.75 g/kg, while the NH_4_^+^-N contents varied from 117.47 to 164.26 mg/kg, which was remarkably higher than the contents of NO_2_^−^-N and NO_3_^−^-N (Table 1).

### 3.2. Diversity and Distribution of Anammox Bacteria in RPAS

Of the nine 16S rRNA gene clone libraries of anammox bacteria, a total of 202 positive clones were identified and separated into 12 OTUs (Figure 2). The positive clones of different samples varied from 17 to 30, with 1–5 OTUs in each sample. A total of 202 sequences from nine libraries yielded 105 unique OTUs, with the highest diversity at the CW in March (CW3). The library coverage values were generally greater than 95.0% except for the PT value in June (PT6), which was 76.5%, indicating that the majority of the anammox bacteria were represented in these clone libraries. The PT6 sample had the highest OTU number, whereas the highest Shannon index (1.21) and lowest Simpson index (0.3) were found in a sample from the CW in September (CW9). The lowest OTU number was detected in the CP in June (CP6) and PT in September (PT9) and only consisted of a single OTU. The highest S_ACE_ estimator was 4.76 in CW3, and the highest Chaol estimator was 11.00 in PT6. Based on the diversity (Shannon and Simpson) and richness indices (S_ACE_ and Chaol) of the16S rRNA genes (Table 2), it shows an apparent unit-specific or seasonal variation in community diversities.

The phylogenetic 16S rRNA gene sequences of the anammox bacteria from this study and their closest relatives are depicted in a consensus tree, as shown in Figure 2. The consensus tree indicated that the 16S rRNA gene sequences are grouped into six clusters, i.e., the Brocadia cluster and A, B, C, D and E clusters, which account for 77.72, 9.41, 0.99, 2.97, 1.98 and 6.93% of all of the sequences selected, respectively. The majority of the 16S rRNA gene sequences (77.72%) belonging to the Brocadia cluster consisted of sequences from all the samples except for that from the CP in June (CP6) (Table 3). These sequences were affiliated with OTU1, 2, 3, 5, 9 and 10 and showed 95.0–100.0% identities with Ca. *Brocadia fulgida* (JX243614), except for OTU-10, which only showed 92.1% identity. OTU-1 was 100% identical to Ca. *Brocadia fulgida*, while OTU-2 and 3 had 98.3 and 97.3% identity, respectively. Cluster E was the third large fraction of sequences (6.93% sequences) and mainly retrieved from PT3 (Figure 2). These sequences shared low identity (72.1–83.1%) with known anammox bacteria and clustered outside all known anammox bacterial genera. In addition, among the twelve OTUs of all samples, OTU-2 and 3 accounted for a percentage greater than 10%, at 44.55% and 25.25%, respectively. This indicated that the sequences related to OTU-2 and 3 represented the most common and dominant anammox bacteria in the aquaculture system. The distributions of anammox bacteria among the samples from the three units presented slight differences (Table 3). Sequences from six of the nine samples (CP3, CP9, PT3, PT6, PT9 and CW3) were completely grouped into the *Brocadia* cluster. All of the sequences from CP6 belonged to cluster A. Sequences from the CW were distributed in all anammox bacteria clusters except for cluster A, indicating that the CW has a comparatively higher anammox community diversity than the CP and PT.

### 3.3. Abundance of Anammox Bacteria in RPAS

The used Real-time quantified PCR analyses showed that the gene copies of the anammox and total bacteria varied with the difference in sampling site (Table 4). Generally, the copy numbers of anammox bacteria varied from 3.33 × 10^1^ to 41.84 × 10^1^ copies per ng of DNA (4.93 × 10^5^–121.80 × 10^5^ copies per g DW for sites A–E), with the highest abundance (41.84 × 10^1^ copies per ng of DNA) observed at site G in June (G6). The abundance of total bacterial 16S rRNA genes ranged from 1.13 × 10^6^ to 7.81 × 10^6^ copies per ng of DNA (3.33 × 10^10^–14.70 × 10^10^ copies per g DW for sites A–E). The highest 16S rRNA gene copies (7.81 × 10^6^ copies per ng of DNA) were detected at site A in June (A6). The proportion of anammox bacteria was between 0.66 × 10^−5^ and 9.42 × 10^−5^, while the proportion in September was generally higher than that in March and June except for site G. In June, the proportion of anammox bacteria in the CW (sites F and G) was higher than that in the other units (CP and PT) (Figure 3).

### 3.4. Correlations of Anammox Bacteria with Environmental Factors

Pearson’s moment correlation analysis indicated that the Shannon index and SACE estimator were significantly (*p* < 0.05) negatively correlated with the NH^4+^-N concentration in water, while Simpson index was positively (*p* < 0.05) correlated with it (Table 5). The abundance of anammox bacteria 16S rRNA genes was significantly *(p* < 0.05) negatively correlated with the NO_2_^−^-N content of sediment. RDA was used to correlate the community assemblages of anammox bacteria with the environmental factors for further elucidation of the varying anammox bacteria community compositions in the aquaculture system. The first two RDA dimensions could explain 81.5% of the total variance in the anammox bacterial composition and the cumulative variance of the anammox bacteria environment relationship (Figure 4).

## 4. Discussion

Numerous previous studies have demonstrated the widespread occurrence of anammox bacteria in natural and man-made ecosystems [10,11,12]. The anammox bacteria genera Ca. Brocadia, Ca. Jettenia and Ca. Kuenenia were reported to be the most common and dominant genera in terrestrial freshwater habitats [25]. This study showed that 77.72% of the sequences retrieved from the RPAS belonged to the Brocadia cluster, suggesting that Ca. Brocadia is the representative anammox bacteria in the RPAS. Ca. Brocadia may be ubiquitous in freshwater ecosystems [26,27]. Ca. Brocadia may have a diverse metabolism and possess strong adaptability [28]. Bryant et al. suggested that more homogeneous organic matter in an environment reduces the various niches available for microbes and therefore leads to decreased diversity [29]. The higher diversity of the CW, compared to that of the other units (CP and PT) observed in this study, is important for optimizing RPAS.

Furthermore, it was estimated that the anammox process accounted for 9–40% of the nitrogen loss in freshwater ecosystems. The proportion of anammox bacteria observed in this study was within the range found in other freshwater [30] and marine [9] environments (10^−5^–10^−3^) but was relatively low. The relatively low proportion observed and the shift in September may be related to some aquaculture environment characteristics. These results showed a heterogeneous spatial and temporal distribution of the anammox bacterial abundance in the RPAS. It is helpful for us to regulate the system environment to give rein to the role of anaerobic ammonia oxidation.

In this study, the NH_4_^+^-N concentration in water was significantly negatively correlated with the Shannon index and S_ACE_ estimator, and the NO_2_^−^-N content in sediment was significantly negatively correlated with the anammox bacteria abundance (Table 5 and Figure 4). High NH_4_^+^-N concentration was also reported to stimulate anammox bacteria in many anammox bioreactors, and relatively high ammonia loading can be beneficial for anammox bacteria growth [17]. The abundance of anammox communities that was both spatially and temporally heterogeneous, obtained here may be related to some specific characteristics in RPAS systems (e.g., recirculating water). This provides a clue for enhancing the anaerobic ammonia oxidation function in wetlands.

In summary, this research makes us more familiar with the anammox bacteria in RPAS, and provides a strong evidence for the regulation of the RPAS.

## Figures and Tables

**Figure 1 ijerph-18-04941-f001:**
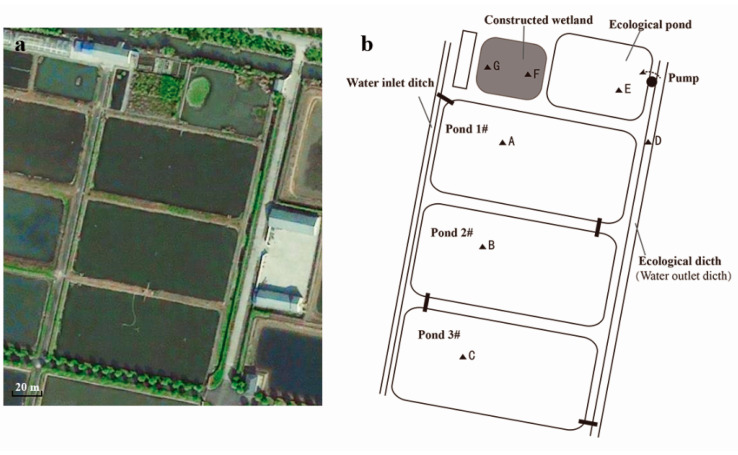
The RPAS and detailed description: (**a**) actual picture of the RPAS; (**b**) structural layout of the RPAS.

**Figure 2 ijerph-18-04941-f002:**
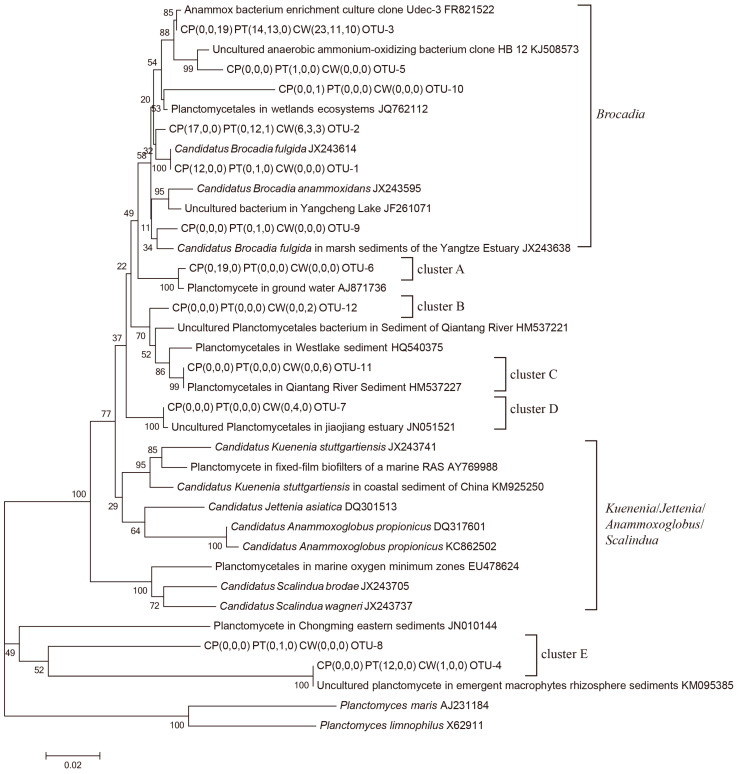
The 16S rRNA gene clone libraries of anammox bacteria in the RPAS.

**Figure 3 ijerph-18-04941-f003:**
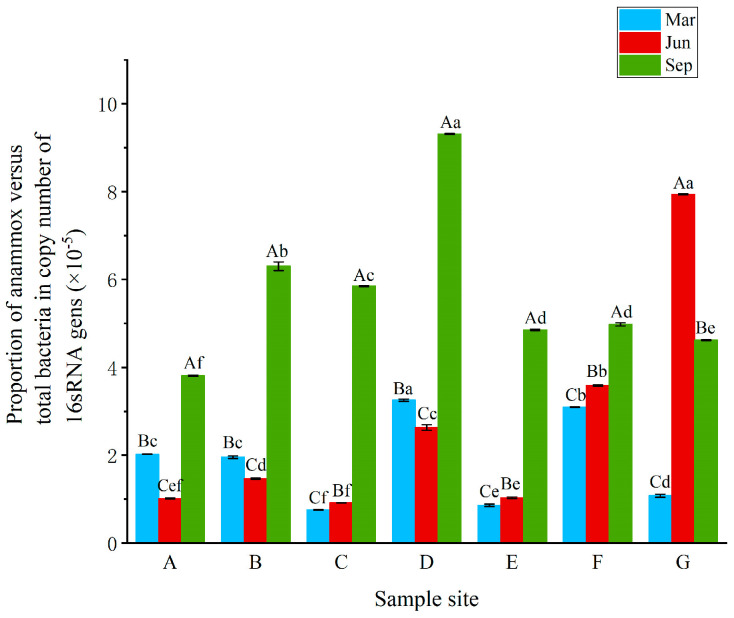
The relative abundances of the anammox bacteria in the RPAS. Letters: a significant diffrerence.

**Figure 4 ijerph-18-04941-f004:**
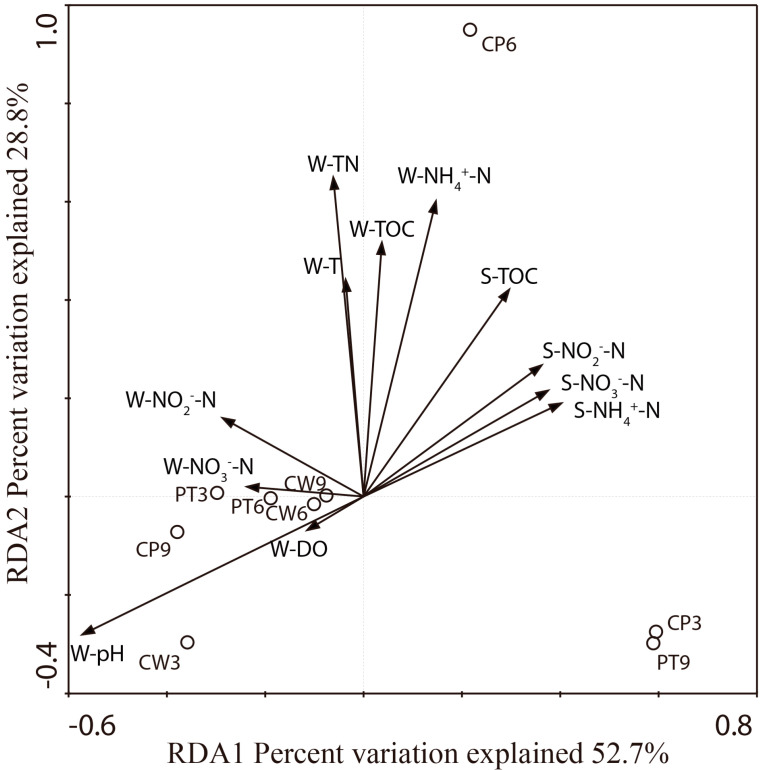
RDA of anammox bacteria and environment relationship.

**Table 1 ijerph-18-04941-t001:** Physicochemical characteristics of water and sediment samples from the RPAS.

Time	Units	Overlying Water/Interstitial Water ^a^								Sediments			
		T (℃)	DO (mg/L)	pH	TOC (mg/L)	TN (mg/L)	NH_4_^+^-N (mg/L)	NO_2_^−^-N (mg/L)	NO_3_^−^-N (mg/L)	TOC (g/kg)	NH_4_^+^-N (mg/kg)	NO_2_^−^-N (mg/kg)	NO_3_^−^-N (mg/kg)
March	CP	11.42 ± 0.02	2.47 ± 0.43	7.02 ± 0.08	nd	nd	nd	nd	nd	2.87 ± 0.31	117.47 ± 23.50	0.19 ± 0.01	4.21 ± 0.42
	PT	13.43 ± 0.35	8.94 ± 0.68	7.77 ± 0.16	nd	nd	nd	nd	nd	3.53 ± 0.72	120.75 ± 31.20	0.13 ± 0.01	3.93 ± 0.99
	CW	11.15 ± 0.21	1.95 ± 0.78	7.52 ± 0.11	nd	nd	nd	nd	nd	–	–	–	–
June	CP	24.87 ± 0.23	2.54 ± 0.61	7.20 ± 0.17	11.17 ± 0.29	16.44 ± 1.93	0.79 ± 0.02	0.03 ± 0.00	0.11 ± 0.03	3.75 ± 0.37	136.77 ± 17.19	0.18 ± 0.02	3.63 ± 0.42
	PT	27.10 ± 0.21	6.06 ± 1.35	7.60 ± 0.05	10.60 ± 0.30	14.89 ± 2.19	0.50 ± 0.02	0.02 ± 0.00	0.04 ± 0.01	2.93 ± 0.19	143.54 ± 22.70	0.19 ± 0.04	2.30 ± 0.28
	CW	24.44 ± 0.69	1.74 ± 0.83	7.65 ± 0.04	8.92 ± 0.19	13.46 ± 1.39	0.33 ± 0.03	0.02 ± 0.01	0.19 ± 0.10	–	–	–	–
September	CP	23.35 ± 0.02	2.93 ± 0.04	7.54 ± 0.05	8.62 ± 0.33	9.24 ± 0.69	0.58 ± 0.03	0.10 ± 0.00	0.56 ± 0.02	2.16 ± 0.06	128.50 ± 40.53	0.16 ± 0.01	3.57 ± 0.49
	PT	22.58 ± 0.18	5.03 ± 1.88	7.63 ± 0.11	8.16 ± 0.19	5.71 ± 1.54	0.55 ± 0.02	0.02 ± 0.00	0.17 ± 0.02	2.20 ± 0.08	164.26 ± 75.35	0.13 ± 0.00	2.93 ± 0.24
	CW	23.20 ± 0.85	1.07 ± 0.49	7.50 ± 0.08	6.97 ± 0.14	6.45 ± 0.57	0.21 ± 0.05	0.02 ± 0.01	0.17 ± 0.03	–	–	–	–

^a^ Overlying water for the CP and PT and interstitial water for the CW; “nd” no data, “–” not applicable; “CP” culture pond unit, “PT” pre-treatment unit, “CW” constructed wetland unit. Values are the mean ± SD calculated from three samples for the CP and two samples for the PT and CW, respectively.

**Table 2 ijerph-18-04941-t002:** Biodiversity and predicted richness of the anammox 16S rRNA genes recovered from the RPAS.

Time	Units	Clones	Unique Sequences	OTUs	Coverage (%)	Shannon	Simpson	S_ACE_	Chaol
March	CP	29	9	2	100.0	0.68	0.50	0.00	2.00
	PT	27	14	3	96.3	0.82	0.45	0.00	3.00
	CW	30	23	3	96.7	0.64	0.62	4.76	3.00
June	CP	19	9	1	100.0	0.00	1.00	0.00	1.00
	PT	17	10	5	76.5	0.87	0.57	0.00	11.00
	CW	18	15	3	100.0	0.93	0.42	3.00	3.00
September	CP	20	13	2	95.0	0.20	0.90	0.00	2.00
	PT	21	7	1	100.0	0.00	1.00	0.00	1.00
	CW	21	16	4	100.0	1.21	0.30	4.00	4.00

“CP” culture pond unit, “PT” pre-treatment unit, “CW” constructed wetland unit.

**Table 3 ijerph-18-04941-t003:** Relative abundance of the phylogenetic types based on the anammox 16S rRNA genes recovered from the RPAS.

	Total Numbers		Relative Abundance (%)								
			March			June			September		
	OTUs	Clones	CP	PT	CW	CP	PT	CW	CP	PT	CW
*Brocadia*	6	157	100	100	100	0	100	77.78	100	100	61.9
Cluster A	1	19	0	0	0	100	0	0	0	0	0
Cluster B	1	2	0	0	0	0	0	0	0	0	9.52
Cluster C	1	6	0	0	0	0	0	0	0	0	28.57
Cluster D	1	4	0	0	0	0	0	22.22	0	0	0

“CP” culture pond unit, “PT” pre-treatment unit, “CW” constructed wetland unit.

**Table 4 ijerph-18-04941-t004:** Quantification of anammox and total bacterial 16S rRNA gene copies in the RPAS.

		March						
		A	B	C	D	E	F	G
Abundance (copies/ng DNA)	Anammox Bacteria (×10^1^)	10.02 ± 0.66ac	9.45 ± 4.13ac	4.04 ± 1.82a	18.83 ± 4.18b	3.89 ± 0.43a	11.31 ± 2.96c	4.18 ± 2.22a
	Total Bacteria (×10^6^)	4.81 ± 0.10a	4.81 ± 0.47a	6.11 ± 0.44b	5.89 ± 0.75bc	4.86 ± 0.53b	3.61 ± 0.45c	3.69 ± 0.39c
Abundance (copies/g DW)	Anammox Bacteria (×10^5^)	17.76 ± 1.16a	15.29 ± 6.68a	7.57 ± 3.42b	22.11 ± 4.91a	4.93 ± 0.54b	–	–
	Total Bacteria (×10^10^)	8.52 ± 0.18a	7.78 ± 0.76ad	11.45 ± 0.82c	6.91 ± 0.88bd	6.15 ± 0.67b	–	–
		**June**						
		**A**	**B**	**C**	**D**	**E**	**F**	**G**
Abundance (copies/ng DNA)	Anammox Bacteria (×10^1^)	8.31 ± 0.92ab	7.96 ± 1.21ab	3.33 ± 0.05a	11.16 ± 2.92ab	3.66 ± 2.33a	16.83 ± 4.09b	41.84 ± 9.46c
	Total Bacteria (×10^6^)	7.81 ± 0.12a	5.41 ± 0.34b	3.33 ± 0.09c	4.35 ± 0.15d	3.58 ± 0.22c	4.61 ± 0.39d	5.19 ± 0.57b
Abundance (copies/g DW)	Anammox Bacteria (×10^5^)	23.88 ± 2.63a	13.15 ± 2.00b	5.53 ± 0.08b	34.94 ± 9.15c	9.20 ± 5.85b	–	–
	Total Bacteria (×10^10^)	22.44 ± 0.33a	8.94 ± 0.56b	5.53 ± 0.15c	13.61 ± 0.46d	9.00 ± 0.56b	–	–
		**September**						
		**A**	**B**	**C**	**D**	**E**	**F**	**G**
Abundance (copies/ng DNA)	Anammox Bacteria (×10^1^)	4.32 ± 0.61a	19.54 ± 4.55b	14.08 ± 4.00bc	19.53 ± 0.20b	10.08 ± 2.83ac	15.82 ± 3.77ab	13.11 ± 4.65ab
	Total Bacteria (×10^6^)	1.13 ± 0.00a	3.12 ± 0.20b	2.41 ± 0.29c	2.07 ± 0.28c	2.06 ± 0.11c	3.10 ± 0.26b	2.82 ± 0.17b
Abundance (copies/g DW)	Anammox Bacteria (×10^5^)	49.86 ± 7.06a	83.24 ± 19.38a	85.86 ± 24.42a	121.80 ± 1.22b	48.32 ± 13.58a	–	–
	Total Bacteria (×10^10^)	13.07 ± 0.06a	13.28 ± 0.86a	14.70 ± 1.75a	12.93 ± 1.73a	9.89 ± 0.51b	–	–

Sites A, B and C are in the culture pond unit (CP); sites D and E are in the pre-treatment unit (PT); and sites F and G are in the constructed wetland unit (CW). Significant differences (*p* < 0.05) between the sites at an identical time are indicated by different lowercase letters. Values are the mean ± SD calculated from the triplicate determination; “–” not applicable.

**Table 5 ijerph-18-04941-t005:** Statistical analysis of anammox community structures with the physicochemical parameters in the recirculating pond aquaculture system (RPAS).

Physicochemical Parameters	OTUs	Shannon	Simpson	S_ACE_	Chaol	Abundance
W-Temperature	0.124	−0.136	0.274	−0.178	0.325	0.314
W-DO	0.132	−0.048	0.043	−0.585	0.292	−0.223
W-pH	0.384	0.226	−0.152	0.168	0.280	0.446
W-TOC	−0.075	−0.350	0.394	−0.619	0.278	−0.455
W-TN	0.129	−0.024	0.046	−0.266	0.326	−0.175
W-NH_4_^+^-N	−0.642	**−0.876 ***	**0.900 ***	**−0.872 ***	−0.267	−0.597
W-NO_2_^−^-N	−0.267	−0.377	0.387	−0.357	−0.264	−0.152
W-NO_3_^−^-N	−0.308	−0.278	0.256	−0.118	−0.425	0.140
S-TOC	0.113	0.267	−0.307	0.000	0.055	−0.680
S-NH_4_^+^-N	−0.155	−0.513	0.628	0.000	0.090	0.396
S-NO_2_^−^-N	0.300	0.233	−0.170	0.000	0.416	**−0.855 ***
S-NO_3_^−^-N	−0.491	−0.007	−0.217	0.000	−0.699	−0.069

***** Correlation is significant at the 0.05 level (two-tailed). Bold: Significant difference (*p* < 0.05).

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
