# Peer review of "Anaerobic Ammonium Oxidation Bacteria in a Freshwater Recirculating Pond Aquaculture System"

_ijerph, 2021, doi:10.3390/ijerph18094941_

Round 1

Reviewer 1 Report

Thank you for the MS with an interesting subject. Unfortunately, otherwise well-performed study suffered from poor english language, especially in the beginning of the MS. I strongly advise to use language check by a native speaker.

Reviewer 2 Report

The work presented by Liu et al. studies the variation of anammox bacterial communities through different spatial and temporal stages in a freshwater recirculating pond aquaculture system, which is very interesting. However, the material and methods section needs to be more accurate and the results section requires further phylogenetic analyses.

Broad comments

There are several English mistakes throughout the text, which sometimes make the reading very difficult to follow. Please, improve this point.

It’s necessary to include more details on how the PCR amplification, qPCR and cloning were performed and which primers were used (i.e. names, their sequences, references).

Provide more details on how the bacterial identification was performed (i.e phylogenetic analyses, software employed).

The phylogenetic analysis needs further elaboration.

Specific comments

Lane 2 Title: "in a freshwater"

Line 11: ". But there is"

Line 12: "In this paper"

Lines 13-15: This sentence is difficult to follow. Please clarify it.

Lines 18-19: Which parameter? Please, clarify this point.

Lines 36-40: This sentence is too long. Please, split it up.

Line 43: "efficiency" is redundant here, please remove it.

Line 43: "improves survival rate" seems to be an additional advantage. Please, clarify it.

Lines 47-48: The meaning of anammox should be placed at the beginning of the section

Line 48: "However"

Lines 48-50: This sentence is difficult to understand. Please, clarify it.

Line 61 and Figure1B: To better follow the results section please provide these acronyms within the Figure 1B.

Line 62: This reference is not listed in the references section.

Lines 65-71: How were the water samples collected? Were they filtered? Please, provide more details on this.

Line 80: Nested PCR or Quantitative PCR?

Lines 80-87: Please provide further details on the methods performed (i.e. cloning, qPCR, and sequencing). Provide the PCR amplification conditions and also the primer sequences.

Were the 16S rRNA genes cloned and sequenced? If yes how? Please, clarify it.

Lines 88-92: Please, provide the details on what phylogenetic analyses were carried out.

Line 89: Provide a reference to this program.

Line 90: Why is this reference (Sogin et al.) cited here?

Line 95: “period usually 10 months” What does this mean?

Lines 100-101: Please, provide here an appropriate reference with the concentration values of these chemicals that are suitable for fishing.

Line 125, Figure 2 legend: Please, explain what the meaning of the numbers on the branches and in brackets. If the first are bootstrap values provide the number of replicates. The confidence for the placement of some clusters are very weak. Please provide additional phylogenetic analyses with an outgroup.

Line 136: Please, write scientific names in italics here and throughout the text.

Line 168 Figure 3: This figure is not cited in the text. Please, clarify or remove it.

Line 169 Figure 4: Indicate the meaning of DW.

Lines 210-211: This sentence is difficult to understand.

Round 2

Reviewer 1 Report

Thank you for revisioning the MS. All my comments and questions were answered and the changes were made accordingly. In my opinion, the MS can now be published.

Author Response

R: Thank you very much for your review.

Reviewer 2 Report

The revised version of the work presented by Liu et al. has not been improved according to previous comments.

Overall, I still have the following concerns about this paper:

Broad comments

- There are several English mistakes throughout the text, which sometimes make the reading very difficult to follow. Please, improve this point.

- It’s necessary to include more details on how the PCR amplification, qPCR and cloning were performed and which primers were used (i.e. names, their sequences, references).

- Provide more details on how the bacterial identification was performed (i.e phylogenetic analyses, software employed).

- The phylogenetic analysis needs further elaboration.

Specific comments

Lane 2 Title: "in a freshwater"

Line 11: ". But there is"

Line 12: "In this paper"

Lines 13-15: This sentence is difficult to follow. Please clarify it.

Lines 18-19: Which parameter? Please, clarify this point.

Lines 36-40. This sentence is too long. Please, split it up.

Line 43: "efficiency" is redundant here, please remove it.

Line 43: "improves survival rate" seems to be an additional advantage. Please, clarify it.

Line 48: "However"

Lines 48-50: This sentence is difficult to understand. Please, clarify it.

Line 61 and Figure1B:To better follow the results section please provide these acronyms within the Figure 1B.

Line 62: This reference is not listed in the references section.

Lines 65-71: How were the water samples collected? Were they filtered?. Please, provide more details on this.

Line 80: Nested PCR or Quantitative PCR?

Lines 80-87: Please provide further details on the methods performed (i.e. cloning, qPCR, and sequencing). Provide the PCR amplification conditions and also the primer sequences.

Were the 16S rRNA genes cloned and sequenced? If yes how? Please, clarify it.

Lines 88-92: Please, provide the details on what phylogenetic analyses were carried out.

Line 89: Provide a reference to this program.

Line 90: Why is this reference (Sogin et al.) cited here?

Line 95: “period usually 10 months” What does this mean?

Lines 100-101: Please, provide here an appropriate reference with the concentration values of these chemicals that are suitable for fishing.

Line 125, Figure 2 legend: Please, explain what the meaning of the numbers on the branches and in brackets. If the first are bootstrap values provide the number of replicates. The confidence for the placement of some clusters are very weak. Please provide additional phylogenetic analyses with an outgroup.

Line 136: Please, write scientific names in italics here and throughout the text.

Line 168 Figure 3: This figure is not cited in the text. Please, clarify or remove it.

Line 169 Figure 4: Indicate the meaning of DW.

Lines 210-211: This sentence is difficult to understand.

Author Response

Lane 2 Title: "in a freshwater"

R Thank you very much for your good suggestions. We has revised.

Line 11: ". But there is"

R We have deleted this section from your good suggestions

Line 12: "In this paper"

R We have deleted this section from your good suggestions

Lines 13-15: This sentence is difficult to follow. Please clarify it.

R Thank you very much, we has revised

Lines 18-19: Which parameter? Please, clarify this point.

R Thank you very much, we has revised

Lines 36-40: This sentence is too long. Please, split it up.

R Thank you very much, we has revised

Line 43: "efficiency" is redundant here, please remove it.

R Thank you very much, we has revised

Line 43: "improves survival rate" seems to be an additional advantage. Please, clarify it.

R Thank you very much, we has revised

Lines 47-48: The meaning of anammox should be placed at the beginning of the section

R Thank you very much, we has revised

Line 48: "However"

R Thank you very much, we has revised

Lines 48-50: This sentence is difficult to understand. Please, clarify it.

R Thank you very much, we has revised

Line 61 and Figure1B: To better follow the results section please provide these acronyms within the Figure 1B.

R Thank you very much, we has revised

Line 62: This reference is not listed in the references section.

R Thank you very much, we has supplement

Lines 65-71: How were the water samples collected? Were they filtered? Please, provide more details on this.

R Thank you very much, we has supplement

Line 80: Nested PCR or Quantitative PCR?

R Thank you very much, we has revised. It is quantitative PCR

Lines 80-87: Please provide further details on the methods performed (i.e. cloning, qPCR, and sequencing). Provide the PCR amplification conditions and also the primer sequences.

R Thank you very much, we has supplement

Were the 16S rRNA genes cloned and sequenced? If yes how? Please, clarify it.

R Thank you very much, we has supplement

Lines 88-92: Please, provide the details on what phylogenetic analyses were carried out.

R Thank you very much, we has supplement

Line 89: Provide a reference to this program.

R Thank you very much, we has supplement

Line 90: Why is this reference (Sogin et al.) cited here?

R Thank you very much, we has revised

Line 95: “period usually 10 months” What does this mean?

R Thank you very much, we has revised

Lines 100-101: Please, provide here an appropriate reference with the concentration values of these chemicals that are suitable for fishing.

R Thank you very much, we has revised

Line 125, Figure 2 legend: Please, explain what the meaning of the numbers on the branches and in brackets. If the first are bootstrap values provide the number of replicates. The confidence for the placement of some clusters are very weak. Please provide additional phylogenetic analyses with an out group.

R Thank you very much, we has revised

Line 136: Please, write scientific names in italics here and throughout the text.

R Thank you very much, we has revised

Line 168 Figure 3: This figure is not cited in the text. Please, clarify or remove it.

R Thank you very much, we has revised

Line 169 Figure 4: Indicate the meaning of DW.

R Thank you very much, we has revised

Lines 210-211: This sentence is difficult to understand.

R Thank you very much, we has revised
